# Nanobody-Based EGFR-Targeting Immunotoxins for Colorectal Cancer Treatment

**DOI:** 10.3390/biom13071042

**Published:** 2023-06-26

**Authors:** Javier Narbona, Luisa Hernández-Baraza, Rubén G. Gordo, Laura Sanz, Javier Lacadena

**Affiliations:** 1Department of Biochemistry and Molecular Biology, Faculty of Chemical Sciences, Complutense University, 28040 Madrid, Spain; jnarbona@ucm.es (J.N.); luisa.hernandez@ulpgc.es (L.H.-B.); rugarc09@ucm.es (R.G.G.); 2University Institute of Biomedical and Health Research (IUIBS), Las Palmas University, 35016 Las Palmas de Gran Canaria, Spain; 3Molecular Immunology Unit, Biomedical Research Institute, Hospital Universitario Puerta de Hierro, Majadahonda, 28222 Madrid, Spain; lsalcober@salud.madrid.org

**Keywords:** immunotoxin, nanobody, antibody engineering, colorectal cancer, antitumor efficacy, α-sarcin

## Abstract

Immunotoxins (ITXs) are chimeric molecules that combine the specificity of a targeting domain, usually derived from an antibody, and the cytotoxic potency of a toxin, leading to the selective death of tumor cells. However, several issues must be addressed and optimized in order to use ITXs as therapeutic tools, such as the selection of a suitable tumor-associated antigen (TAA), high tumor penetration and retention, low kidney elimination, or low immunogenicity of foreign proteins. To this end, we produced and characterized several ITX designs, using a nanobody against EGFR (V_HH_ 7D12) as the targeting domain. First, we generated a nanoITX, combining V_HH_ 7D12 and the fungal ribotoxin α-sarcin (αS) as the toxic moiety (V_HH_EGFRαS). Then, we incorporated a trimerization domain (TIE^XVIII^) into the construct, obtaining a trimeric nanoITX (TriV_HH_EGFRαS). Finally, we designed and characterized a bispecific ITX, combining the V_HH_ 7D12 and the scFv against GPA33 as targeting domains, and a deimmunized (DI) variant of α-sarcin (BsITXαSDI). The results confirm the therapeutic potential of α-sarcin-based nanoITXs. The incorporation of nanobodies as target domains improves their therapeutic use due to their lower molecular size and binding features. The enhanced avidity and toxic load in the trimeric nanoITX and the combination of two different target domains in the bispecific nanoITX allow for increased antitumor effectiveness.

## 1. Introduction

Colorectal cancer currently occupies the third position in incidence (1.9 million new cases per year) and second place in terms of mortality (935.000 annual deaths) among the different cancer types [1]. Current therapies consist mainly of surgery, followed by chemotherapy based on fluoropyrimidines [2] or radiotherapy [3] as adjuvant treatments. However, these are non-specific therapies, leading to off-target toxicities and directly affecting the quality of life of patients. In this context, immunotherapy and the advent of monoclonal antibodies have arisen as a breakthrough in the treatment of this disease, as they are targeted therapies with great specificity and promising clinical results [4,5].

With regard to passive immunotherapy based on monoclonal antibodies, immunotoxins (ITXs) are a type of immunoconjugate that can be used in the treatment of both tumoral and infectious diseases [6,7]. ITXs are chimeric molecules with a target domain, usually an antibody fragment such as Fab (fragment antigen-binding) or scFv (single-chain variable fragment), but also cytokines or growth factors that specifically direct the action of a toxic protein of bacterial or plant origin towards the cancerous cell, leading to its death [8].

As target domains, nanobodies (V_HH_) are the single variable domains of heavy-chain-only antibodies obtained from camelids [9]. These nanobodies retain the full antigen-binding potential and are considered the smallest naturally-derived antigen-binding fragments. Nanobodies exhibit several advantages over full-length antibodies or scFvs, such as smaller size (~15 kDa), allowing deeper penetration in solid tumors, the ability to cross the blood–brain barrier [10], or their increased stability and resistance to high temperatures (60–80 °C) and pressures (500–700 MPa), extreme pHs (3.0–9.0), and chemical denaturants, such as 2–3 M guanidinium chloride or 6–8 M urea [11,12]. In addition, they exhibit a longer complementary determining region loop 3 (CDR3), which enables nanobodies to bind with high affinity to traditionally inaccessible epitopes [13,14]. Regarding immunogenicity, nanobodies are mostly considered non-immunogenic proteins, since they do not trigger either T-cell proliferation or anti-V_HH_ antibody production, possibly due to the lack of Fc fragments and the high homology sequence with human V_H_ domains [15]. All these features have led to the medical use of nanobodies in imaging and treatment applications, including the treatment of viral infections such as HIV or SARS-CoV2 [16,17], or several cancer types [18,19].

The main critical issue for the therapeutic effect of immunoconjugates and ITXs is the selection of target tumor-associated antigens (TAAs) [8,20]. Ideal TAAs should be expressed only in tumoral tissue and be absent from healthy cells to avoid off-target toxicities. In addition, the internalization rate of the receptor after ITX binding may be a more determinant step than the receptor expression level [21,22]. In this sense, nanoITXs targeting HER2 [23], the transferrin receptor [24], IL2R [25], or mesothelin [26], among others, have already been described.

GPA33 is a transmembrane glycoprotein that is overexpressed in 95% of primary and metastatic colorectal cancers, while it is mostly absent in healthy tissues [27,28]. It is believed to be involved in cell adhesion functions and the development of the intestinal mucosa during the embryonic early stages [29]. GPA33 remains constitutively attached to the membrane cell, showing a high degree of internalization after the binding of the corresponding antibody, and is thus an ideal TAA for targeting [30].

EGFR, also named ErbB1, is a tyrosine kinase receptor that belongs to a family of four receptors: ErbB1 (EGFR), ErbB2 (HER2), ErbB3 (HER3), and ErbB4 (HER4). They are transmembrane glycoproteins that upon the binding of their ligand, EGF or TGF-α, homo- or hetero-dimerize and activate downstream signaling routes, such as RAS/RAF/MEK/ERK, PI3K/AKT, and JAK/STAT3, involved in the proliferation and survival of a great number of different cell types [31,32]. Because of these functions, the overexpression of EGFR confers great advantages on tumoral cells in epithelial cancers, such as in 15–30% of breast cancers, 60% of lung cancers, and 80% of colorectal cancers [33].

Regarding the toxic domain, toxins from different bacteria and plants have been routinely used, such as *Pseudomonas* exotoxin A, *Diphtheria* toxin, gelonin, and the plant toxin ricin, among others [34,35]. However, they present some drawbacks that limit their use as therapeutic drugs, such as immunogenicity. These toxins trigger an immune response from the patient, who thereby develops anti-drug antibodies (ADAs). These ADAs neutralize the immunotoxin, preventing further treatment cycles [36,37].

Ribotoxins, a protein family that belongs to the fungal extracellular RNase superfamily, have emerged as promising toxic domains due to their small size, high thermostability and resistance to proteases, low immunogenicity, and efficient cytotoxic activity. Ribotoxins recognize a conserved motif in the large rRNA subunit, namely the sarcin/ricin loop (SRL), and produce a single cleavage, leading to protein biosynthesis inhibition and cell death by apoptosis [38,39]. Among ribotoxins, the most characterized member is α-sarcin, a small basic ribotoxin of 150 amino acids that has been employed as the toxic domain in the design of several ITXs against allergies [40] and cancers, showing potent antitumoral activity both in vitro and in vivo [41,42,43,44].

In order to overcome the versatility and extreme adaptability of colorectal cancer, we produced and characterized several designs of ITXs, based on the V_HH_ 7D12 [45] that binds to the EGFR extracellular domain III, preventing its dimerization and subsequent signaling activity and thus taking advantage of the unique features of nanobodies as the target domain. First, we designed the nanoITX V_HH_EGFRαS, combining the V_HH_7D12 and α-sarcin as the toxic domain. Second, we designed a trimeric version, TriV_HH_EGFRαS, to increase the avidity and toxic load of the nanoITX. In addition, its larger size should avoid fast renal clearance, leading to increased treatment efficacy. TriV_HH_EGFRαS includes the collagen XVIII trimerization domain (TIE^XVIII^) flanked by 18 amino acid linkers, leading to a trimeric nanoITX [46,47,48]. Finally, we produced a bispecific immunotoxin, BsITXαSDI, combining the V_HH_ 7D12 and the scFvA33 against GPA33 [49,50], along with a non-immunogenic variant of α-sarcin (αSDI) [51]. This bispecific design allows the recognition of two different TAAs and therefore would maintain its efficacy in case of antigen loss during tumor progression.

## 2. Materials and Methods

### 2.1. Plasmid Design

Plasmids containing the cDNA sequence encoding V_HH_ 7D12 and TIE^XVIII^ domains were provided by Laura Sanz’s group from Hospital Puerta del Hierro. cDNA corresponding to scFvA33, α-sarcin, and αSDI were previously obtained by our group. The desired cDNA sequences were amplified by PCR, including the restriction sites needed for cloning, and then cloned in pPICZαA (Invitrogen, Carlsbad, CA, USA), to finally obtain the plasmids pPICZαAV_HH_EGFRαS, pPICZαATriV_HH_EGFRαS, and pPICZαABsITXαSDI. All three plasmids contained a Zeocin resistance gene, and the plasmid construction included an α-factor signal at the N-terminal site, to allow extracellular secretion, and a six-histidine tag at the C-terminal site, to allow their purification. The three plasmids were sequenced by the Genomic Unit at the Universidad Complutense de Madrid, to confirm the desired sequences.

### 2.2. Protein Production and Purification

Electrocompetent *Pichia pastoris* KM71H strain cells were electroporated with 10 μg of linearized plasmid after digestion with the restriction enzyme *Pme*I, using a Bio-Rad Gene Pulser device. After the electroporation pulse, the cells were seeded in YPDS plates containing increasing zeocin concentrations (100 to 750 μg/mL). Multiple clones were selected to test the optimal conditions for their expression, modifying the induction time (24–72 h) and temperature (15–25 °C). Protein expression was analyzed using 0.1% (*w*/*v*) sodium dodecyl sulfate (SDS)–15% polyacrylamide gel electrophoresis (PAGE) and Western blot using an anti-α-sarcin antibody.

Once the optimal conditions were selected for each nanoITX, large-scale expression was carried out. First, the selected yeast clone was grown in 2 L of BMGY medium using baffled Erlenmeyer flasks, at 30 °C with vigorous agitation. After 24 h, the cells were harvested by centrifugation, and resuspended in 1 L of BMMY for induction, inducing the protein expression at the previously optimized temperature and time. Once the induction was finished, the extracellular medium was dialyzed several times against 50 mM sodium phosphate, 0.1 M NaCl, and pH 7.5 buffer.

The three nanoITXs were purified by following the same protocol, including affinity chromatography using a Ni^2+^-NTA agarose column (GE Healthcare). First, the medium was applied to the column at a flow rate of 1 mL/min. Then, the column was washed with 50 mM sodium phosphate, 0.1 M NaCl, and pH 7.5 buffer, and then washed again with the same buffer containing 20 mM imidazole. The nanoITXs were eluted by the addition of the same buffer containing 250 mM imidazole. Fractions containing the desired protein were pooled, and dialyzed against a sodium phosphate buffer, to remove the imidazole.

### 2.3. Structural Characterization

Structural characterization was carried out by several means. Absorbance measurements were taken using a UV-1800 spectrophotometer device (Shimadzu, Shimadzu Europa GmbH, Duisburg, Germany). Secondary structure information was obtained from the record of far-UV circular dichroism (CD) spectra, using a Jasco J-715 spectropolarimeter (Jasco Analítica, Madrid, Spain). NanoITX samples were resuspended in 50 mM sodium phosphate buffer, 0.1 M NaCl, and pH 7.5 at a final concentration of 0.2 mg/mL. Cells with an optical path of 0.1 cm were employed, and eight spectra were averaged to obtain the final spectra. To evaluate the thermal stability of the nanoITXs, thermal denaturation profiles (Tm) were obtained by measuring the temperature dependence of the molar ellipticity at 220 nm of a 0.2 mg/mL nanoITX solution in the 20–85 °C temperature range, with a temperature increase rate of 30 °C/hour. FPLC was performed in an AKTA purifier device (GE Healthcare Lifescience) using a Superdex 200 column to analyze the trimeric format and the molecular size in the solution of the trimeric nanoITX (TriV_HH_EGFRαS).

### 2.4. Ribonucleolytic Activity Assays

The highly specific ribonucleolytic activity of α-sarcin or its deimmunized variant, α-sarcin DI (αSDI), was assayed, as previously described [52,53], against ribosomes from a rabbit cell-free reticulocyte lysate. The subsequent release of the characteristic 400 nt rRNA fragment, known as α-fragment, was used to confirm the ribonucleolytic activity of α-sarcin. Briefly, a rabbit cell-free reticulocyte lysate (50 μL) was incubated with different amounts of nanoITXs. The ribonucleolytic reaction was stopped by the addition of 250 μL of Tris 50 mM, SDS 5%, and pH 7.5 buffer; RNA was isolated with a phenol:chloroform:isoamyl alcohol (25:24:1) extraction, and then precipitated by the addition of isopropanol. The RNA pellet was washed with −20 °C ethanol 70% and resuspended in 10 μL of DEPC H_2_O. α-Fragment presence was visualized by electrophoresis in a 2% agarose, 16% paraformaldehyde gel, prestained with ethidium bromide. Images were obtained using a Universal Hood II Transilluminator device.

### 2.5. Cell Line Cultures

The A431 cell line (ATCC CRL-1555, Rockville MD, USA) was used as a tumoral EGFR+ epidermoid carcinoma cell line [54], whereas the SW1222 cell line (ATCC HB-11028, Rockville, MD, USA) was used as a GPA33+ and EGFR- colorectal cancer cell line. Both cell lines were cultured in RPMI medium, supplemented with 300 mg/mL of L-glutamine, 50 μg/mL of penicillin, 50 mg/mL of streptomycin, and 10% of fetal bovine serum. Cells were cultured at 37 °C in a humidified atmosphere (CO_2_:air, 1:19 *v:v*). Harvesting and propagation of both cell lines were performed each 2–3 days through trypsinization, and the number of cells was calculated with a Neubauer chamber.

### 2.6. ELISA Assay

The binding ability of the nanoITXs towards its antigen, EGFR, was first analyzed by ELISA. Plates were coated with EGFR (0.5 µg/mL) at 4 °C, overnight. Then, after three phosphate-buffered saline (PBS) washes, the wells were blocked with BSA 5% PBS, and incubated with the different nanoITXs (1 μM) for 1 h at RT. After three more PBS washes, an anti-Histag-HRP secondary antibody diluted 1/2000 was added to each well and incubated for 2 h at RT. After incubation, the wells were washed again with PBS and 100 μL of substrate solution was added. The reaction was stopped by the addition of 100 μL of H_2_SO_4_ 2M. The V_HH_EGFR nanobody was used as a positive binding control, and BSA-coated wells were used as a negative control. Triplicates of each condition were tested. The results, represented as mean ± standard deviation, were calculated as the subtraction of the absorbance at 450 nm of EGFR-coated wells from those of the BSA-coated wells.

### 2.7. Flow Cytometry Studies

Trypsinized cells were distributed into aliquots of 3 × 10^5^ cells/mL and washed three times with 300 μL of BSA 1% (*w/v*) PBS. Then, cells were incubated with different concentrations of nanoITXs for 45 min at room temperature with gentle agitation. After another three washes with BSA 1% (*w/v*) PBS, a second incubation was carried out, using a diluted 1/100 anti-Histag Alexa 488 antibody (Sigma Aldrich, St. Louis, MO, USA) for 45 min at room temperature with gentle agitation. Then, the cells were finally washed three times with BSA 1% (*w/v*) PBS, and fluorescence was measured using a FACScan (Becton Dickinson, NJ, USA), at the Centro de Apoyo a la Investigación of the Universidad Complutense de Madrid. The results were analyzed using the FlowJo software (FlowJo v10, Oregon, OR, USA).

### 2.8. MTT Viability Assay

To analyze the cytotoxicity of the nanoITXs, MTT viability assays were performed. Cells were trypsinized and seeded into 96-well plates at a cellular density according to the cell line, in a culture medium, and maintained under standard culture conditions for 24 h. Then, different concentrations of immunotoxin diluted in an FBS-free medium were added to the cells and kept in culture conditions for 24, 48, and 72 h. After the incubation, each well was incubated with 20 μL of MTT (5 mg/mL) for 4 h. Then, 100 μL of DMSO:methanol (1:1, *v/v*) was added to each well, to dissolve the formazan crystals formed by the reduction of MTT carried out by the enzymatic activity of live cells. The results were obtained by colorimetric quantification, measuring the optical density at 570 nm and expressed as a percentage of viability. Triplicates of each condition were tested. Medium-cultured cells were used as the 100% viability control and the IC_50_ was determined as the ITX concentration that led to a 50% decrease in the cell viability.

### 2.9. In Vivo Antitumor Assay

All animal procedures were carried out according to the guidelines of the Universidad Complutense Animal Experimentation Committee, and the Community of Madrid official regulations (Royal Decree 53/2013). Balb/c nude male mice (7 weeks old) were purchased from Harlan Laboratories (Barcelona, Spain) to analyze the in vivo effect of V_HH_EGFRαS against human epidermoid cancer xenografts. Assays were performed at the Animal Facility of the Centro de Investigaciones Biológicas-Consejo Superior de Investigaciones Científicas (CIB-CSIC) in Madrid.

Mice were split into three experimental groups (n = 5): PBS (phosphate-buffered saline) and V_HH_EGFRαS 25 or V_HH_EGFRαS 50, according to the dose administered (25 or 50 μg of nanoITX per injection). Prior to the experimental procedure, the animals were given a 7-day adaptation period with free access to food and water. Each mouse received a subcutaneous injection into the right flank of 2 × 10^6^ A431 cells, resuspended in 200 μL of 1:1(*v/v*) PBS–Matrigel (BD Biosciences, San Jose, CA, USA) mixture. Once the tumor volume reached 50–100 mm^3^, the mice were injected intravenously either with PBS or different doses of V_HH_EGFRαS. Five doses, either of PBS or of the two doses (25–50 μg) of V_HH_EGFRαS were given every 48 h. Tumors were measured each 48 h using an external caliper, and volume was calculated with the formula: tumoral volume = length × width^2^ × 0.52. After the administration of 5 doses (day 10), the drug administration stopped, but the tumoral volume measurement was continued until the volume was too high (2500 mm^3^) and the animals had to be sacrificed. Survival analysis of the experimental mice was carried out using the Kaplan–Meier representation. ANOVA with a post hoc analysis using the Student–Newman–Keuls test was used for statistical analysis within each test to compare the results obtained with the different doses administered. All values were expressed as arithmetic means ± sem (standard error of the mean). Differences between experimental groups were considered statistically significant at *p* < 0.05.

## 3. Results

### 3.1. Generation, Production, and Purification of NanoITXs

The expression vectors for the production of the three nanoITX variants (Figure 1) were electroporated into KM71H Pichia pastoris cells, and the proteins were successfully secreted to the extracellular medium by the addition of methanol.

The three nanoITXs were purified from the extracellular medium by immobilized metal affinity chromatography and were analyzed by SDS-PAGE electrophoresis, followed by Coomassie blue staining or Western blot immunodetection using an anti-α-sarcin antibody (Figure 2). V_HH_EGFRαS and BsITXαSDI showed an expected mass of 34 and 58 kDa, respectively, showing a high degree of purity. TriV_HH_EGFRαS purification resulted in two different bands, 45 and 30 kDa, with only the 45 kDa band reacting with the anti-α-sarcin serum (Figure 2e). The purification yields of V_HH_EGFRαS, TriV_HH_EGFRαS, and BsITXαSDI were around 4, 0.5, and 3 mg/L of induction media, respectively.

### 3.2. Structural Characterization

All nanoITX circular dichroism (CD) spectra were compatible with water-soluble globular proteins with a high degree of β-sheet secondary structure (Figure 3a–c). TriV_HH_EGFRαS spectra showed a high percentage of random structure, due to the contribution of the 18 residue linkers that flank both sides of the trimerization motif and the TIE^XVIII^ domain inserted between the target and the toxic domain in the trimeric nanoITX. FPLC size exclusion chromatography was carried out in order to confirm the trimeric conformation of TriV_HH_EGFRαS in native conditions (Figure 3e). The size exclusion chromatography elution profile (Figure 3e) showed a main peak at an elution volume corresponding to 130 KDa, as expected for the trimeric ITX. The thermal denaturation profiles of V_HH_EGFRαS and BsITXαSDI showed Tms of 50 and 55 °C, respectively (Figure 3d,e), indicating that both immunotoxins are highly stable at high temperatures, which is consistent with the high stability of both the nanobodies and α-sarcin.

### 3.3. Ribonucleolytic Activity

Functional characterization of the three nanoITXs was first carried out by analyzing the function of each domain separately. In all functional assays, the concentration of TriV_HH_EGFRαS was calculated as a monomer, to highlight the advantages of the trimeric over the monomeric format. The ribonucleolytic activity of the toxic domain was assayed using the reticulocyte assay as described in Materials and Methods, by the specific release of the α-fragment from the larger subunit of the rRNA. The release of the α-fragment was observed with all nanoITXs demonstrating that ribonucleolytic α-sarcin activity was preserved (Figure 4); this was also the case with non-immunogenic α-sarcin (αSDI) (Figure 4b), showing small differences depending on the amount of protein assayed, within 80–100% of the activity compared to wild-type α-sarcin (Figure 4c).

### 3.4. Binding Activity

First, an ELISA using immobilized EGFR was carried out, in order to confirm the correct binding of V_HH_EGFRαS, TriV_HH_EGFRαS, and BsITXαSDI to its antigen. As observed in Figure 5a, the three nanoITXs were able to bind to immobilized EGFR, in a similar way as the positive control (V_HH_EGFR), retaining at least 80% of the binding ability of the control. These differences can be explained in terms of possible steric impairments due to the presence of the toxic domain or the second target domain. In addition, in order to consider the native conformation of EGFR in a cellular context, flow cytometry assays were carried out, using the EGFR+ A431 cell line. V_HH_EGFRαS was able to bind specifically to A431 cell lines, in a dose-dependent manner (Figure 5b).

Flow cytometry assays using A431 cells incubated with TriV_HH_EGFRαS showed that the trimeric ITX recognizes and binds A431 cells more efficiently than its monomeric counterpart (Figure 5c), reaching binding saturation at the low concentration of 10 nM (Figure 5d). Moreover, BsITXαSDI (Figure 5e,f) was able to bind to both SW1222 (GPA33+, EGFR-) and A431 cells (EGFR+, GPA33-), matching the binding obtained for its monospecific counterparts, scFvA33 for SW1222 cells [41] and V_HH_EGFRαS for A431 cells (Figure 5b).

### 3.5. In Vitro Antitumoral Activity

To assess their therapeutic potential, the in vitro antitumoral effect of the different nanoITXs was analyzed. Tumor cells were incubated with different concentrations of nanoITXs for different periods of time, and viability was measured using the MTT viability assay. V_HH_EGFRαS exhibited a cytotoxic effect on A431 cells, reducing the viability of tumoral cells in a dose- and time-dependent manner, showing an IC_50_ of 300 nm at 72 h (Figure 6a). Interestingly, TriV_HH_EGFRαS showed a significantly higher cytotoxic effect on the same cells compared to its monomeric counterpart, with an IC_50_ at 72 h of 10 nm (Figure 6b), thirty times lower than V_HH_EGFRαS.

Regarding BsITXαSDI, the bispecific immunotoxin was cytotoxic against both SW1222 and A431 cells, with IC_50_ values at 72 h of 700 nM and 1 μM, respectively (Figure 6c,d). The IC_50_ values of BsITXαSDI were higher than those obtained by the monospecific immunotoxins against the respective antigen-positive cell line, with an IC_50_ of 300 nM for V_HH_EGFRαS and 700 nm for IMTXA33αS [41].

### 3.6. In Vivo Antitumoral Activity

Finally, the in vivo antitumor activity of V_HH_EGFRαS was studied, employing for this purpose nude mice bearing A431 (EGFR+) epidermoid cancer xenografts. Two different doses of V_HH_EGFRαS were assayed (25 or 50 μg) along with a negative control (PBS) (Figure 7a). Treatment with V_HH_EGFRαS led to significant inhibition of tumor growth (three times in the 25 μg group and four times in the 50 μg group) (Figure 7c). Remarkably, we observed a medium-lasting tumor growth rebound after the last injection, reaching a final volume of 2000 mm^3^ at day 25.

These data on tumor volume were in agreement with the survival analysis by Kaplan–Meier representation. In Figure 7b, it is shown that the survival rate of the mice treated with 25 μg of V_HH_EGFRαS at day 27 was 40% and rose to 80% with the 50 μg dose treatment at the end of the experiment.

## 4. Discussion

Chimeric immunotoxins are promising therapeutic tools that have demonstrated promising activity in pre-clinical and clinical studies, against hematological and solid tumors [55], infectious diseases [6], and autoimmune processes [56]. However, efficacy, stability, tumor penetration, and immunogenicity are still aspects that must be improved, both in the target and the toxic domain of the immunotoxin design, to enable more extended therapeutic use.

To this end, in this work, we successfully designed and characterized different formats of nanoITXs based on the nanobody 7D12, specific against the extracellular domain of EGFR [57]. The binding affinity (Kd) of V_HH_ 7D12 for EGFR was found to be in the range of 219 to 279 nM [58]. Its advantages in epitope recognition, compared to different Fab and complete antibodies directed against the same tumor marker, have been also described. Thus, this nanobody became a very interesting alternative for obtaining different multivalent or multispecific therapeutic designs [58]. First, the monomeric nanoITX, V_HH_EGFRαS, consisting of the 7D12 nanobody as the targeting domain and the fungal ribotoxin α-sarcin as the toxic domain, was produced. Then, we designed a trimeric format, TriV_HH_EGFRαS, introducing a collagen-XVIII-derived trimerization domain, TIE^XVIII^, between the target and the toxic domain. Finally, a bispecific immunotoxin (BsITXαSDI) was generated by combining the nanobody anti-EGFR with an scFv against GPA33 and a non-immunogenic variant of α-sarcin.

V_HH_EGFRαS was successfully produced in the yeast *P. pastoris*, with higher yields than those obtained for other ITXs based on scFv possibly due to its smaller size [41,59]. The structural characterization of all nanoITXs showed a high proportion of β-sheet secondary structure, which is consistent with the secondary structure of their components, the V_HH_ 7D12 [60] and the ribotoxin α-sarcin [61,62]. The Tm of V_HH_EGFRαS was 50 °C, lower than that obtained for the V_HH_ alone, which is around 80 °C [19,63], suggesting that the incorporation of α-sarcin could diminish the stability of the construction. However, the Tm at 50 °C is high above the physiological temperature of 37 °C, at which the V_HH_EGFRαS should exert its therapeutic effect.

The functional characterization of V_HH_EGFRαS showed that both the toxic and the target domains were able to exert their ribonucleolytic activity and specific binding to EGFR, respectively. The in vitro cytotoxic activity against the A431 cell line resulted in an IC_50_ of 300 nM at 72 h, slightly higher than for other nanoITXs described against EGFR [64]. This lower cytotoxic efficacy could be due to a lower internalization rate of the nanoITX, or to different issues regarding the release of the toxic domain once it has been internalized. Inclusion of the furin site in the linker between both domains could be achieved, according to the results described for other ITXs [44]. V_HH_EGFRαS showed potent inhibition of tumoral growth in A431-cell-xenografted mice, comparable to other immunotoxins whose in vivo antitumor activity has already been characterized [42]. Tumor growth rebound observed once treatment has been completed could be explained by lower tumor retention of the nanoITX, due to its monovalence or to the fast glomerular clearance of V_HH_EGFRαS, since its molecular weight (35 kDa) is lower than the glomerular filtration threshold [65,66].

The design of new formats of ITXs, thanks to the development of antibody engineering technology, has made it possible to overcome these drawbacks through the construction of multimeric and multispecific formats, with improvements in biodistribution, serum-half life, and tumoral penetration and retention [43,67]. In this sense, we designed a trimeric nanoITX, TriV_HH_EGFRαS, by the incorporation of a collagen-XVIII-derived trimerization domain (TIE^XVIII^) between the target and the toxic domain. After its production in *P. pastoris*, the electrophoretic analysis showed two different bands: a 45 kDa band that was recognized by the anti-α-sarcin antibody, and would consequently correspond to a monomer of TriV_HH_EGFRαS, and a 30 kDa band, not recognized by the anti-α-sarcin antibody, and that would presumably correspond to a monomer of TriV_HH_EGFR, without the toxic domain, thus indicating that it was soon cleaved before the whole protein was translated and secreted to the extracellular medium. The structural characterization showed that TriV_HH_EGFRαS presented a higher content of random structure than its monomeric counterpart, due to the flexible loops that flank the trimerization domain [46]. Size-exclusion chromatography showed that TriV_HH_EGFRαS eluted in a major peak, with the expected molecular size of 130 kDa, confirming its trimeric format in solution. The functional characterization of TriV_HH_EGFRαS showed that it reached binding saturation at a concentration of 10 nM, indicating better targeting activity than that of its monomeric counterpart. This improvement in targeting activity was also observed in the evaluation of the cell viability of both nanoITXs, with improved cytotoxic activity observed for the trimeric format, leading to a 30-fold lower IC_50_ compared to the monomeric format. Both results indicate that the trimeric format represents an advance in the design of therapeutic antibodies, as assessed with other trimeric immunotoxins [43]. It has been postulated that the binding of one monomer of the target domain to its antigen could enhance the binding of the other targeting domains. This increase in avidity is facilitated by the flexible 21 amino acid linker that flanks the trimerization domain [68,69]. Moreover, the enhanced binding activity, together with the release of three toxin molecules to the cytosol, could explain an increase in the overall cytotoxic activity, with a significant decrease in the IC_50_, and could potentially allow a reduction in the amount of drug administered, with the same therapeutic effect and minimizing the potential adverse effects. Most importantly, the molecular size of TriV_HH_EGFRαS, far above the renal cut-off, is expected to increase the half-life and decrease the frequency of dosing along with improved tumor retention [48]. In this sense, in vivo experiments with TriV_HH_EGFRαS are yet to be carried out.

Although anti-EGFR mAb cetuximab and panitumumab have improved the clinical response and overall survival of metastatic colorectal cancer patients [70], several unmet needs remain. Patients that initially responded to cetuximab treatment became refractory to the treatment because of downstream signaling mutations, such as mutations in KRAS or PIK3CA [67,71]. One of the most notable mutations of EGFR, namely EGFRvIII, consists of the deletion of exons 2–7, resulting in a truncated extracellular domain that lacks the ligand binding site, and gains constitutive kinase activity [72]. In addition, intrinsic heterogeneity and tumor evolution allow tumors to escape monospecific therapeutics, due to antigen loss [73,74]. In this sense, antibody technology has allowed the design of multispecific antibodies, that can bind to different epitopes in the same antigen [75], to different antigens on the same cell type, or to different antigens on different cell types, allowing the recruitment of immune cells to the tumoral environment [37,75].

In this sense, we designed and produced a bispecific immunotoxin (BsITXαSDI) that presents V_HH_7D12 against EGFR and the scFv against GPA33 as the target domains, and a non-immunogenic variant of α-sarcin (αSDI) as the toxic domain [51]. BsITXαSDI was successfully produced and purified to homogeneity in *P. pastoris*, with a final yield of 3 mg/L of induction, indicating that, as in the case of V_HH_EGFRαS, the presence of the V_HH_ 7D12 in the N-terminus leads to a higher rate of production.

The structural characterization showed that BsITXαSDI presents a correct folding as a water-soluble globular protein, with a high content of β-sheet structure, as expected given the structure of its two target domains and the contribution of the αSDI. BsITXαSDI exhibited structural stability at temperatures above the physiological threshold of 37 ºC.

The toxic domain of BsITXαSDI, the non-immunogenic α-sarcin variant, kept its ribonucleolytic activity, as shown by the release of the α-fragment in the reticulocyte assay. The results obtained in the binding assays by flow cytometry confirmed that both target domains were functional. Thus, BsITXαSDI, under saturated concentration conditions was able to bind to both the EGFR+ A431 and the GPA33+ SW1222 cell lines, in a similar manner to its monospecific counterparts, V_HH_EGFRαS and the previously described scFvA33αS [50]. The IC_50_ values of BsITXαSDI for both cell lines were higher than those obtained with the monospecific variants, suggesting a lower internalization rate, possibly due to the higher molecular weight of the bispecific immunotoxin. However, several studies have shown that bispecific antibodies showing lower affinities for their TAAs are more likely to have better safety profiles since the binding of both targeting domains is required for an efficient intracellular uptake, whereas high-affinity targeting domains could bind to non-malignant cells expressing only a single TAA in their membrane [36,55]. In this sense, the dual targeting of two different TAAs could be an effective tool for overcoming primary tumor heterogeneity, and subsequent tumor escape due to antigen loss, which makes tumors refractory to conventional treatments. In addition, the inclusion of αSDI allows a safer and more effective therapeutic tool since it has proven to be effective both in vitro and in vivo [76] and may prevent the formation of potentially neutralizing anti-drug antibodies (ADAs) by the patient´s immune system.

## 5. Conclusions

The results obtained from the characterization of the three nanobody-based immunotoxins confirm and support their huge potential in the diagnosis and treatment of cancer. The special features of nanobodies, including high binding affinity, thermostability, and low immunogenicity, are of special interest in the diagnosis and treatment of several cancer types, including hematological tumors [77], glioblastoma [78], and colorectal cancer [79,80,81]. In addition, due to their modular arrangement, it is possible to design new immunotoxins, incorporating different functional domains with multimeric formats or different specificities, that endow these immunotoxins with optimized features against cancer.

The work herein described constitutes an effort to overcome the challenges of treating a heterogeneous disease such as colorectal cancer. The nanoITXs here described benefit from the smaller size of nanobodies and their excellent binding properties allowing deeper tumor penetration. The trimeric ITX improves the overall antitumor efficacy due to its increased avidity and toxic payload. Finally, the combination of multiple target domains directed against different tumoral antigens, combined with the absence of immunogenicity derived from the αSDI, represents a step forward in the treatment of this disease. The next steps will involve the study of the in vivo antitumor efficacy of the trimeric nanoITX, as well as the effect of combined treatments, and also the design of new optimized bispecific formats, using fungal ribotoxin-based nanoITXs as a platform for the design of new therapeutic constructs.

## Figures and Tables

**Figure 1 biomolecules-13-01042-f001:**
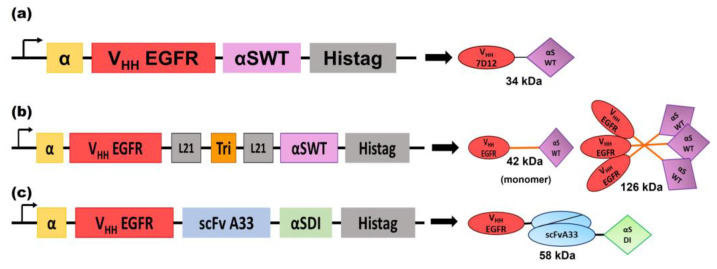
Schematic diagrams showing the genetic and protein domain arrangements of the nanoITXs V_HH_EGFRαS (**a**), TriV_HH_EGFRαS (**b**), and BsITXαSDI (**c**). The cDNA constructions appear on the left side of the figure. The schematic representation of the protein motifs of the nanoITXs appears on the right side of the figure, with its molecular size underneath. In both cases, structural and functional domains are highlighted with different colors: α-factor secretion signal peptide (α, yellow), V_HH_EGFR (corresponds to V_HH_ 7D12 [45]) (red), scFvA33 (blue), the 21 aa flexible linkers (gray), the trimerization domain derived from the collagen XVIII (TIE^VIII^, orange), the wild-type α-sarcin (αSWT, purple), the non-immunogenic α-sarcin (αSDI, green), and histidine-tag (Histag, black).

**Figure 2 biomolecules-13-01042-f002:**
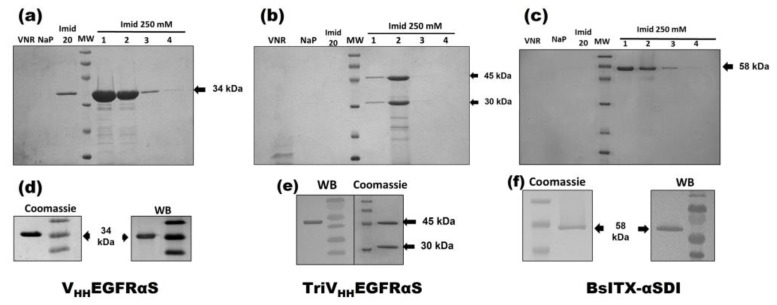
Coomassie-blue-stained SDS-PAGE analysis after affinity purification of V_HH_EGFRαS (**a**), TriV_HH_EGFRαS (**b**), and BsITXαSDI (**c**) and Coomassie blue staining and Western blot analysis (WB) of the purified final fraction of V_HH_EGFRαS (**d**), TriV_HH_EGFRαS (**e**), and BsITXαSDI (**f**). Western blot analysis was carried out using rabbit anti-α-serum. Notes in gels correspond to the following: MW, prestained molecular weight standard (kDa); VNR, not retained fraction; NaP, washed fraction eluted with sodium phosphate buffer; Imidazole 20 mM, washed fraction eluted with sodium phosphate buffer containing imidazole 20 mM; and different 1 mL fractions eluted with 250 mM imidazole. The original full-length gels and uncropped Western Blot images can be found in Appendix A.

**Figure 3 biomolecules-13-01042-f003:**
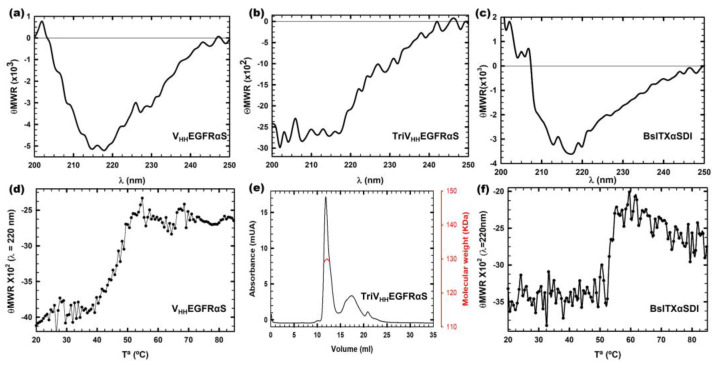
Structural characterization by far-UV circular dichroism (CD) spectra of V_HH_EGFRαS (**a**), TriV_HH_EGFRαS (**b**), and BsITXαSDI (**c**). θMRW represents the mean residue weight ellipticities as degree × cm^2^ × dmol^−1^. The thermal denaturation profiles of V_HH_EGFRαS (**d**) and BsITXαSDI (**f**) by means of the temperature dependence of the ellipticity at 220 nm. All spectra were carried out at a protein concentration of 0.15 mg/mL in 50 mM sodium phosphate, 0.1 M NaCl, and pH 7.4. Analysis of the trimeric nature of TriV_HH_EGFRαS by Superdex 200 FPLC chromatography analysis (**e**). The eluted protein shows a major symmetric elution peak at the expected volume corresponding to its trimeric size (126 kDa), with the indicated molecular weight measured at the center of the chromatography peak (red curve).

**Figure 4 biomolecules-13-01042-f004:**
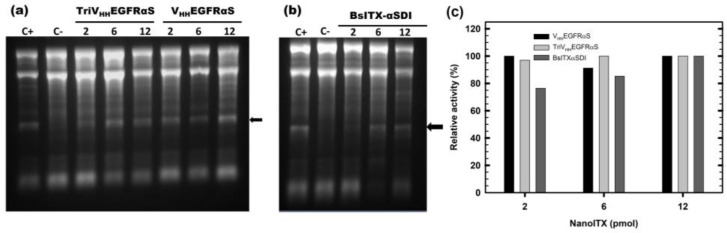
In vitro functional characterization. The ribonucleolytic activity of the toxic domain of V_HH_EGFRαS and TriV_HH_EGFRαS (**a**), and BsITXαSDI (**b**). The arrow indicates the release of the α-fragment, produced by the cleavage of the SRL due to the α-sarcin. In both gels, 2, 6, and 12 pmoles of all three nanoITXs were assayed. C+ represents 2 pmoles of fungal wild-type α-sarcin, whereas in C-, the protein sample was replaced by a buffer. Images were acquired and analyzed using the Gel Doc XR Imaging System and the Quantity One software (BioRad). (**c**) Quantitation of specific ribonucleolytic activity of the three nanoITXs, expressed as a percentage of α-fragment/RNA 18S ratio, considering 100% to be the ratio obtained by 2 pmol of α-sarcin. Band intensities were quantitated with Quantity One software. The original full-length gels can be found in Appendix A.

**Figure 5 biomolecules-13-01042-f005:**
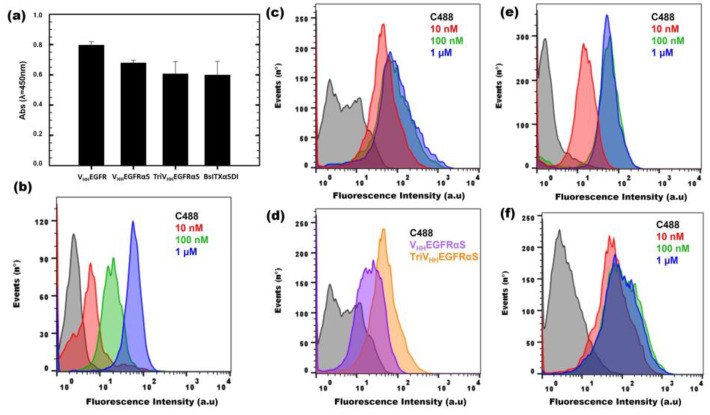
In vitro functional characterization of binding activity. ELISA assay (**a**) against immobilized EGFR (0.5 μg/well), using the three nanoITXs and V_HH_EGFR as a positive control (1 µM). Flow cytometry binding assays of V_HH_EGFRαS (**b**) and TriV_HH_EGFRαS (**c**) to EGFR-positive cells (A431 cell line). Curves correspond to cells incubated with a secondary antibody anti-His-Alexa488 (black), 10 nM (red), 100 nM (green), or 1 μM (blue) of each immunotoxin. (**d**) Comparison between 10 nM of V_HH_EGFRαS (purple) or TriV_HH_EGFRαS (yellow) in the flow cytometry assay. Binding of BsITXαSDI to EGFR-positive cells (A431) (**e**) or GPA33-positive cells (SW1222) (**f**). Curves correspond to cells incubated with a secondary antibody anti-His-Alexa488 (black), 10 nM (red), 100 nM (green), or 1 μM (blue).

**Figure 6 biomolecules-13-01042-f006:**
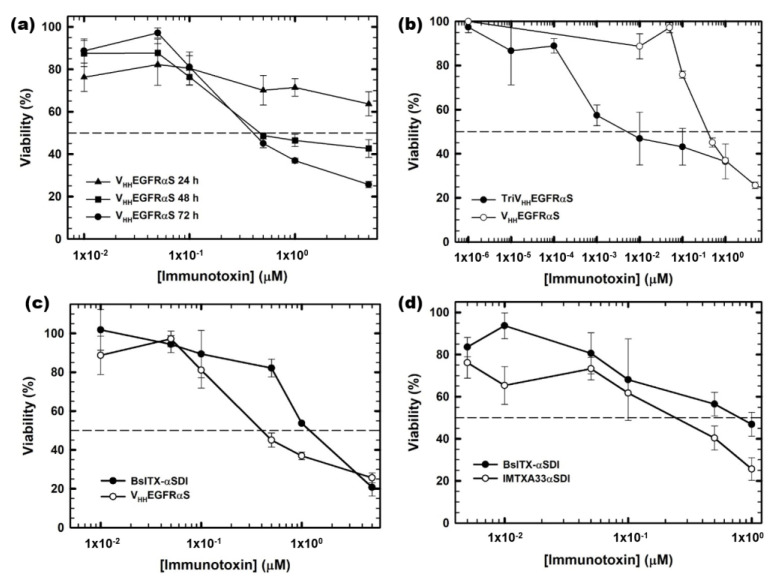
In vitro cytotoxicity characterization by MTT viability assays. (**a**) EGFR-positive A431 cells treated with V_HH_EGFRαS for 24, 48, and 72 h; (**b**) EGFR-positive A431 cells treated for 72 h with TriV_HH_EGFRαS (black) compared to V_HH_EGFRαS (white); (**c**) EGFR-positive A431 cells treated for 72 h with BsITXαSDI (black) compared to V_HH_EGFRαS (white) and (**d**) GPA33-positive cells SW1222 treated with BsITXαSDI (black) and scFv-IMTXA33αSDI (white). Measurements were analyzed and plotted (mean ± SD) against untreated controls. In all cases, triplicate samples were carried out. IC_50_ values were obtained as the protein concentration leading to 50% viability.

**Figure 7 biomolecules-13-01042-f007:**
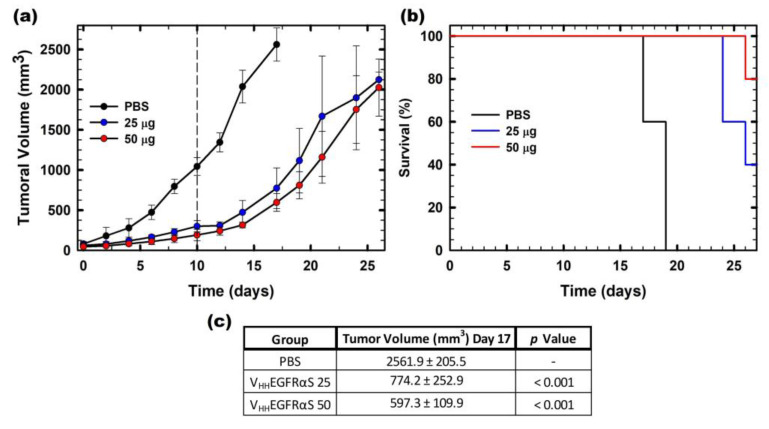
In vivo antitumoral activity. (**a**) Time course of tumor volume growth of SW1222-derived xenografts. Mice were non-treated with PBS (black) or treated with two different doses (25 or 50 µg) of V_HH_EGFRαS (labeled in the graph as 25, blue or 50, red). The different doses were administered each 48 h. The vertical dashed line indicates the end of the administration of the antitumoral treatment. Values are represented as means ± sem (standard error of the mean). (**b**) Kaplan–Meier survival curves. The Kaplan–Meier representation expresses the time to the experimental endpoint (once tumor volume reaches 2000 mm^3^ of the in vivo assay). The labels in the graph are the same as those used in (**a**). (**c**) Statistical analysis of V_HH_EGFRαS 25- and V_HH_EGFRαS 50-treated tumors compared with vehicle-treated tumors at day 17. In all cases, the experimental groups were composed of 5 mice (n = 5).

## Data Availability

All experimental data generated or analyzed during this study are included in the article.

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
