# Peer review of "Nanobody-Based EGFR-Targeting Immunotoxins for Colorectal Cancer Treatment"

_biomolecules, 2023, doi:10.3390/biom13071042_

Round 1

Reviewer 1 Report

This is an interesting protein engineering approach to treat cancer, with early data.

Comments below.

The word “new” in the title isn’t necessary.

Supplementary figures: label MWt markers.

Throughout: the abbreviation for kiloDaltons has a lower case k

Line 252: some figures are Std deviation, which is preferable to SEM.

Line 271: should be (CD)

Line 278: spelling of chromatography

Line 289 should be “reticulocyte assay”

Figure 4: were the gels quantified? Would be good to add that info.

Line 300: change “unspecific” to “non-specific”

Line 310, 311: sentence does not make sense and has spelling error.

Line 333: “wearing” is not the right word here

Figure 3: please put the molecule name on each panel to make it easier for the reader.

Figure 5a: what is C- in (a)?

Figure 5(d) legend change nm to nM

Figure 7. Is tumor volume error SD or SEM?

Line 376: rewrite this sentence

Fix spelling in the title of ref 36

Please provide any information on antibody affinity measurements (Kd).

What are the next steps for this study?

Please review grammar and spelling throughout. This can be easily and cheaply performed with OpenAI or Grammarly.

There are lots of small errors.

Reviewer 2 Report

Dear Editor,

My comments are here below:

The paper describes the design and the characterization of three immunotoxins useful for targeting colon cancer cells. All three immunotoxins are described at functional level“in vitro” model, but only VHHEGFRαS is described in “in vivo” murine model. The trivalent and bispecific immunotoxin constructs are however well discussed in section 4, where is specified that “in vivo experiments with TriVHHEGFRαS are yet to be carried out”.

The introduction of the paper is concise in providing information about the various structural components of the immunotoxins. All experiments have appropriate controls. The results are clearly shown and conclusions are validated by results.

I have few observation, my comments are as follows:

Major points

Why weren’t ELISA experiments conducted on all three immunoconstructs?

Only the Elisa result related to the VHHEGFRαS construct is shown, the Elisa of the other two constructs should also be shown.

Minor points

Line 98 “to overcome the versatility and extreme adaptability of colorectal cancer…” is the only sentence that explains why trivalent and bispecific immunotoxin constructs are designed, the explanations are then described in detail in the “Discussion” section. The purpose of a trivalent versus a monovalent construct is certainly intuitive, but the purpose of a bispecific construct for two tumor antigens is only explained in the discussion section. In my opinion, a hint on the reasons why these two constructs are designed should already be added in the “Introduction” section. 

Line 299 the positive control is named VHH 7D12, in the image of the figure 5a is named VHHEGFR: please use the same name.

Line 289/290 ….as previously described……. Previously in materials and methods or previously as described in another paper?: please cite appropriate references or write as described in materials and method.

Figure 6: describe point d more clearly.

Round 2

Reviewer 1 Report

The sentence beginning on Line 103 is extremely long, convoluted, and confusing. please re-write.

Line 278: now is"The elution profile of the chromatography showed a main peak at an elution volume corresponding to 130 KDa, as expected with its trimeric arrangement."

change to: "The size exclusion chromatography elution profile (Fig. 3e)  showed a main peak at an elution volume corresponding to 130 KDa, as expected for the trimeric toxin"

Fig 3e. What is the red scale? Since this is size exclusion chromatography, the peak height is the amount of protein not the MWt. What is the red curve?

Line 200: fix spelling

Line 254 and legend of  Fig 7 please change "standard error of the media" to "standard error of the mean"

Line 358 change "The binding capacity of VHH 7D12 to EGFR has been previously described in depth, determining its Kd in the nM...[58]"

to "The binding affinity (Kd) of VHH 7D12 for EGFR was found to be in the range of XXXnM to XXX nM...[58]."

Throughout, check you have used "toxic"  and "toxin" correctly. You have used toxic where you mean toxin in multiple places.

Statistical methods: SNK ANOVA uses critical values, not P values. Check that you described the method properly.

Fig legend 5

change "(d) Comparison between 10 nM of VHHEGFRαS (purple) or TriVHHEGFRαS (red)" to be (yellow).

Line 383. "difficulties" is not the right word. Please re-write this sentence

Line 422: replace "should" with  "is expected to"

line 473 should be "glioblastoma"

last line of the article:

Please change the last sentence of the article to a general description of the research direction rather than a prediction of the outcome.

Ref 76 is a duplicate of ref 36

Many grammar and spelling errors remain along with other errors and convoluted and confusing writing. Please get help with this.

Reviewer 2 Report

figure legend 6 

…… Viability assays were also performed for (c) BsITXαSDI (black), against (EGFR+) A431 cells, compared to VHHEGFRαS (white) (c) and against (GPA33-positive) SW1222 cells; comparing BsITXαSDI (black) and to scFv-IMTXA33αSDI (white) (d) for 72 hours.

Check the grammatical puntctuation.
